# Microhabitat accessibility determines peptide substrate degradation by soil microbial community

Carlos Arellano-Caicedo,[1,2] Pelle Ohlsson,[3] Saleh Moradi,[1] Edith C. Hammer[1,4]

**ABSTRACT** Soil pore space, considered the most complex biomaterial that exists, generates a complex environment, that gives rise to a wide variety of properties, such as microbial diversity and carbon storage. Soils contain, at the same time, the largest carbon reservoir on earth and an immense amount of nutrient-limited microbial biomass. The reason why this carbon is not consumed by soil microbes is attributed to the complex nature of soil, which forms a labyrinth where carbon and microbes cannot be in direct contact. In the present study, by using microfluidics, we tested the effect of labyrinth-like structures of decreasing accessibility on the decomposing activity of soil microbial communities from a soil inoculum. The two parameters used to study the effect of microhabitat accessibility were either the turning angle in an array of channel-like pore structures or the fractal order in an array of maze-like pore structures. We found that in both cases, channels and mazes, decreasing accessibility produced a higher peptide substrate degradation. When we analyzed the degradation within the structures, we found that most of the activity is concentrated in the regions of intermediate accessibility. We think that the increased degradation activity in low accessibility mazes might be due to the reduced interactions within the microbial communities which leads to a reduction in competition. Lowered competition allows different communities with a wide range of metabolic strategies to cohabit in the structures, which resulted in a bulk increase of the peptide substrate degradation.

**IMPORTANCE** The role microbes have in the environment is highly influenced by the characteristics of their habitat. Here, we show that a complex habitat enhances the enzymatic activity of a soil microbial inoculum. This might occur due to a reduced competition in complex habitats, which allows a more diverse community to coexist and explore a wider variety of metabolic strategies. The different rates of enzymatic activity in different levels of complexity suggest emergent properties of microbial communities in complex microhabitats which could have important implication for microbial processes, such as soil carbon storage and nutrient cycling.

**KEYWORDS** microhabitat, soil bacteria, organic matter stabilization, soil pore space, physical carbon stabilization, organic matter occlusion, microfluidics, micromodel

**Peer Reviewer** Gao Chen, University of Tennessee at Knoxville, Knoxville, Tennessee, USA

Address correspondence to Carlos Arellano-Caicedo, carlos.arellano@univie.ac.at.

The authors declare no conflict of interest.

See the funding table on p. 15.

Complex habitats are responsible for the wide diversity of microbes in ecosystems (1), ranging from marine ecosystems (2), sea floors (3), and gut microbiota (4) to the big array of microhabitats that are formed within soils (5). In soils, spatiotemporal fluctuating patchy nutrient distribution gives rise to diverse array of habitats, a fragmented aqueous interface, and a barrier to cells and molecule dispersion, which affect the distribution, functions, and diversity of microorganisms (6). This gives the soil emergent properties (7), such as hosting a wide diversity of microorganisms in the bulk soil (8) cohabiting aggregates that may act as independent evolutionary incubators (9). One of the most relevant emergent characteristics that arise from the unique nature of the soil pore space

is its capacity to retain large amounts of carbon buried within its structure (10). The preservation of this carbon underground can last from minutes to decades, and it occurs even though soil microbial biomass is found in a constant state of starvation (11). A tentative explanation to this paradox is that due to the physical complexity of the soil pore space, microbial decomposers, and their potential substrate, are not co-located in space, and time and carbon consumption occur only at short and specific times and locations (12).

To explore the way microhabitat characteristics affect soil microorganisms, several approaches have been adopted, which range from studying intact soil aggregates to a simulation of the pore space in artificial microsystems. The study of intact soil structure is mainly conducted with the help of microcomputed X-ray tomography, which detects the inner spatial properties of aggregates, revealing a maze-like porous system where microbial processes take place (13). From the images obtained, information, such as distribution of air pockets, water interface, and particulate organic matter, can be derived. It has been demonstrated that organic matter turnover was linked to the connectivity and accessibility of the pores to the external part of the aggregate (14), that there are correlations between pore size distribution and organic matter loss (15, 16), and that there is a link between the pore characteristics and the phylogenetic composition of microbial communities that they contain (17). Limitations of this technique, nonetheless, arise when trying to have controllable microenvironmental conditions, real-time measurements of microbial communities, undisturbed sampling, and micrometer scale resolution (18), which constrains the testable hypotheses.

The intrinsic characteristics of soil that limit an in-depth study of its nature can, however, be simulated using artificial models that mimic the inner pore space in a controlled way. An important approach is the use of microfluidics, which is defined as the manipulation of structures and fluids at the micro- and nanoscale (19), to test ecological questions. The use of microfluidics for microbial ecology has revealed mechanisms of chemotaxis (20–23), bacterial motility (20), effects of EPS in the resistance of pore spaces to drought (24), and the way fungi and bacteria interact at the cellular level (25).

Previous microfluidic studies, focused on the influence of pore space physical parameters on microbial growth and their nutrient degradation, showed that fungi and bacteria are affected in different ways by turning angles in microchannels (26) and by the connectivity and fractal order of a pore space maze (27). The turning angle characteristics in long, non-connected microchannels increasingly deviating from straight passage reduced bacterial and fungal growth, as well as enzymatic activity inside the channels. In contrast, when testing microhabitats of different fractal order in space-filling fractal mazes, fungal biomass was reduced, while bacterial biomass and enzymatic activity increased as fractal order increased. The spatial patterns of enzymatic activity inside the highest-order fractals indicated that the highest enzymatic activity occurred in regions of intermediate depth into the mazes, while in the deepest regions (i.e., the least connected, longest paths into the maze), enzymatic activity decreased again. This indicated that the different parameters defining spatial accessibility, in the case of these two studies: channel turning angle and maze fractal order, dissimilarly affect the microbial growth and enzymatic activity.

These previous experiments (26, 27) were performed with the laboratory bacterial strain *Pseudomonas putida* and the fungal strain *Coprinopsis cinerea* under sterile laboratory conditions. It remained open whether these patterns would be generally true and thus similar in other microbial strains or in a whole microbial community as from a natural inoculum. For this purpose, we aimed at testing the effect of these pore space physical parameters (turning angle and turning order in microchannels and fractal order of a maze) on the microbial substrate degradation of a microbial inoculum extracted from a soil sample. In a single strain experiment, it was found that channels with sharp turning angle did not affect the enzymatic activity of a single bacterial strain (26). On the other hand, mazes with low connectivity led to higher enzymatic activity (27). This could be the product of cooperation mediated via quorum sensing or to a reduction in

intraspecific competition that mazes but not channels promote. However, we believed this might not reflect what occurs in nature with complex soil communities. Hence, we expected cooperation in natural microbial consortia to be lower than in a single strain since competition would be higher for nutrients and space. We hypothesized that this would lead to higher substrate enzymatic activity in channels with low turning angle and in simple mazes compared to channels with high turning angle and complex mazes, respectively.

## MATERIALS AND METHODS

### Device design

The design of the microfluidic devices that we have named channel device and fractal device were made in AutoCad 2019, and they consist of an experimental area with six and four treatments, respectively, and a pillar system that served as entrance. The pillar system is formed by pillars of 100 µm in diameter, separated by 100 µm, which soil microbes penetrate the full width of the device before entering the treatment areas.

The channel device consists of dead-end channels of six different geometries ($n$ = 10) with the same internal volume. The channels were randomly distributed in parallel orientation along the device. The parameters assigned to the channels were "Angle" and "Turn order." The angles used were 45°, 90°, and 109°, measured as the deviation from a continued straight line and thus the turning angle an organism in this channel needs to perform (Fig. 1). These angles were selected so that they could represent obtuse, right, and acute angles. Channels of each angle had two types of arrangements, one with an alternated turn order and one with a repeated turn order. Channel types with alternated turn order followed a pattern of alternating right and left bends, while channels with repeated turn order followed a pattern of two right turns followed by two left turns. The channel lengths were adjusted so that every type of channel would contain the same volume (2.42 nL) with a width of 10 µm and a height of 12 µm. Each channel segment was 50 µm long before the next turn. The tortuosity of the channels is indicated in Fig. 1. The channel chip design is the same used in our previous work to study growth and substrate degradation of a bacterial and fungal strains in channels of different turning angles (26).

The fractal device consists of an array of dead-end mazes constructed based on the space filling Hilbert curve (28). Four orders of the Hilbert curve were chosen for the experiments, zero, one, three, and five (Fig. 1), with a normalized internal volume. Each maze fractal number indicates the fractal order of the basic forming unit that makes up the entire maze. For instance, if the fractal order of the maze is 3, it means that it is formed by many order 3 fractals that are repeated until they fill up the space in the maze. It was also included a fractal order 0, so that a completely connected maze could be obtained. The forming units of fractal 0 were the units of order 1 of the Hilbert curve but with one wall less out of the three that forms it. With increasing fractal order, accessibility and connectivity in the mazes decrease, and tortuosity and mean geodesic distance to the inlet increase. The width of the channels within the labyrinths was 10 µm, and the height 12 µm. The labyrinths were randomly distributed along the device ($n$ = 7). The fractal chip design is the same as that used in our previous work to study growth and substrate degradation of bacterial and fungal strains in mazes of different fractal order (26).

In the present work, we used the term low accessibility for channels with high turning angle and repeated turning order in the case of the channel chip and mazes with high fractal order in the case of the fractal chip. On the other hand, high accessibility was attributed to channels of low turning angle and alternated turning order and to mazes with low fractal order.

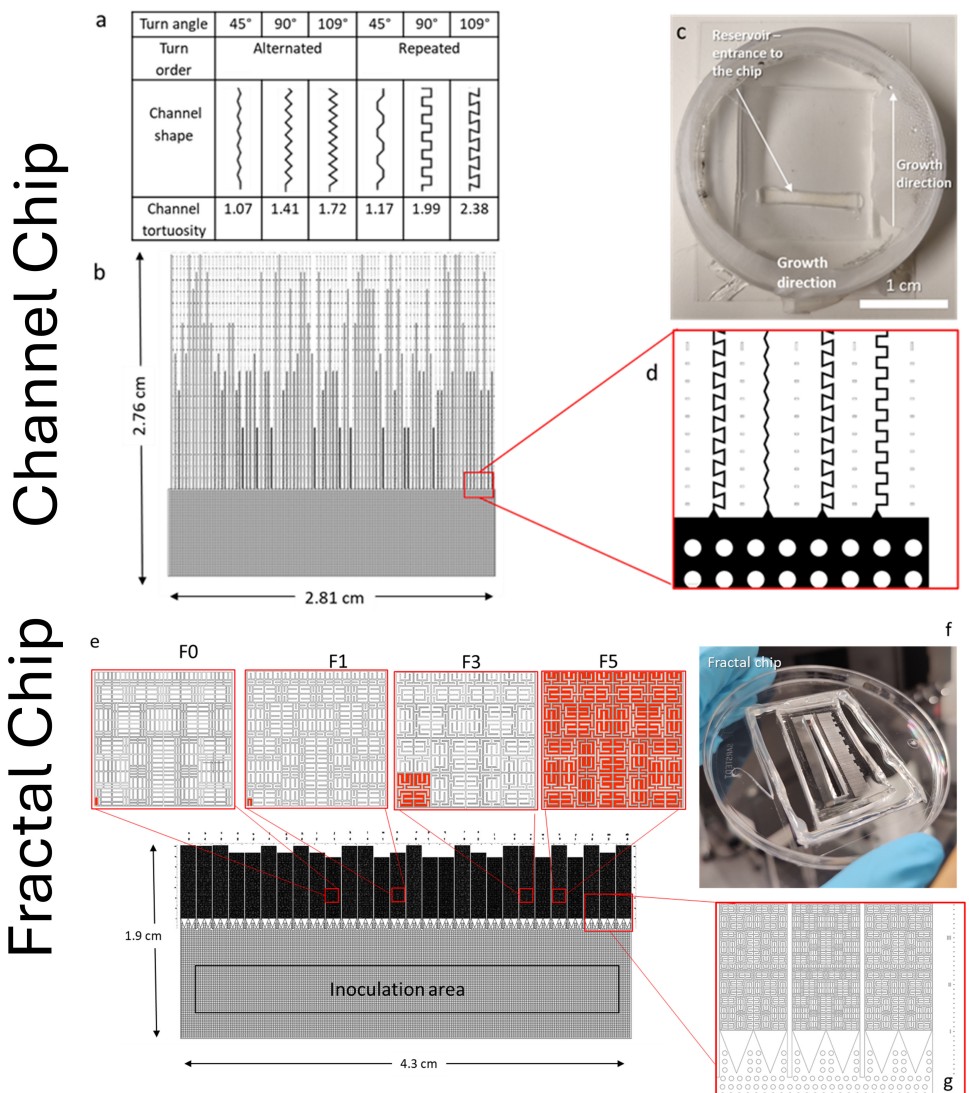

**FIG 1** The two microfluidic devices used in the experiment containing different channels (a–d) and mazes (e–h) (26, 27). The channel device consists of channels with six different conditions: three angles (45°, 90°, and 109°) with two turning orders (alternated, repeated), which resulted in channels with different tortuosities (ratio between arc-chord ratio and the distance between the two ends of the channels). A pillar system connecting to the channels serves as inoculum area. The channel chip illustration is adapted from our previous publication (26), as well as the fractal chip from (27), and is reproduced here for reader convenience (a). The channel device contained 10 internal replicates of each treatment, with a normalized volume, which were distributed randomly along the pillar system (b). The channel device was sterilized and bonded to a glass bottom Petri dish that contained a sterile wet tissue to prevent humidity losses during the experiment. A hole cut into the pillar system area served to enter the inoculum (c). Detailed view of the entries with funnel-shaped connectors to different channels (d). The fractal device contained four different mazes of four fractal orders, 0, 1, 3, and 5, which were named, respectively (F0, F1, F3, and F5). Five maze modules of each fractal order type of the size corresponding to the smallest entity of F5 (as shown in pannel e as red squares) are stacked to blocks that only internally connect to the pillar system. Seven replicate blocks of each maze type, with a normalized internal pore space volume, were randomly located along a pillar system (h). Once molded in PDMS and sterilized, the channel device was bonded to a glass bottom Petri dish that contained a sterile wet tissue to prevent humidity losses during the experiment, with an inoculation reservoir cut into the pillar system (f). Detailed view of the entries to the mazes with funnel-shaped connectors. Each maze block is accessible via three points from the pillar system, and internally connect to the following four maze modules, while a wall separates each block (g).

## Device fabrication

The fabrication of the microfluidic devices was done according to previously described workflows (29). The photomask was made of soda lime glass covered with a thin layer of chromium (Nanofilm, CA, USA). The designed patterns were printed with a dwl66+ mask writer (Heidelberg Instruments, Germany). A Nd:Yag laser, 532 nm, was used to print

the patterns on a photoresist, AZ1500. The patterns were subsequently developed in AZ 351B positive developer and the chromium etched in TechniEtchCr01 (Microchemicals GmbH, Ulm, Germany). For the master fabrication, SU-8 2015 (MicroChem, Newton, MA, USA) was poured onto a heat-dried (90° for 30 minutes) 3-inch silicon wafer (Siegert Wafer, Aachen, Germany) and then spun at 4000 rpm to get a 12µm-thick layer, which determined the height of the device structures. The wafer containing the SU-8 was exposed to UV light in a contact mask aligner (Karl Suss MJB4 soft UV, Munich, Germany). After UV exposure, the non-crosslinked photoresist was developed (MrDev600) and rinsed with isopropanol. To prevent PDMS from sticking to the mold, during the microfluidic device fabrication, the wafer was activated in oxygen plasma for 60 seconds (ZEPTO, Diener Plasma-Surface Technology, Germany) and exposed overnight to a vapor of trichloro (1H,1H,2H,2H-perfluorooctyl) silane (PFOTS, Sigma Aldrich, Saint Louis, MO, USA) at 180 degrees to obtain a monolayer over the structures. SYLGARD 184 PDMS (Dow Chemicals Company, Midland, Michigan) for the microfluidic device fabrication was made by mixing the elastomer base with the curing agent in a proportion of 10:1 in mass, then poured on top of the master that contained the structures, degassed at −15 kPa for 1 hour, and finally polymerized in an oven at 60°C for 2 hours.

PDMS is gas permeable and permits the diffusion of oxygen, water vapor, $CO_2$, and other gasses into the channels (30), meaning that oxygen and other gasses were exchanged with the environment through the chip reservoir and through the PDMS. Together, these factors permit the system not to be oxygen depleted.

The PDMS labyrinths were cut out from the master, and a rectangular portion of 2.5 cm × 0.5 cm for the channel device and 4 cm × 0.5 cm for the fractal device was cut out in the middle of the pillar system, approximately 0.5 cm away from the entrance of every labyrinth. This cut was made to create the reservoir that served as entrance to the labyrinth (Fig. 1). Using a plasma chamber, the PDMS labyrinths and a glass from a glass bottom Petri dish were activated and bonded. This activation consisted in treating the surfaces with a Zepto Plasma System (Diener Plasma Surface Technology, Germany) with these conditions: polarity, negative; coating time, 1 min for cover slips and 10 seconds for PDMS labyrinths. Directly after activation, the surfaces were put together, forming a tight irreversible bonding (31). Directly after bonding, 150 µL and 300 µL of the treatment medium were introduced through the reservoir of the channel and the fractal device, respectively.

## Soil inoculum and growth conditions

The soil used for the experiment was obtained from a grassland of pH 6.5 and a SOM content of 7.9% in weight, located outside the Ecology building of the Lund University, 55° 42′ 49.5′′ N, 13° 12′ 32.5′′ E. One gram of soil was mixed with 20 mL of distilled water and vortexed for 3 minutes at 3,200 rpm. The mixture was allowed to sediment for 5 minutes to let sand and coarse silts collect at the bottom. 1.5 mL of the supernatant was collected and centrifuged for 10 minutes at 5000 RPM to concentrate the microbial extract. This strongly increases microbial cell numbers and diversity in the inoculum and thus decreases the risk of fast-growing species quickly outcompeting most others. The supernatant with the water solution was disposed, and the pellet was resuspended with 100 µL of M9 minimal medium (12.8 g/L NaHPO$_4$·7H$_2$O, 3 g/L KH$_2$PO$_4$, 0.5 g/L NaCl, 100 mg/L NH$_4$Cl, 0.12 g/L MgSO$_4$, 4 g/L d-Glucose, 11.66 mg/L CaCl$_2$, 13.5 mg/L FeCl$_2$, 125 mg/L MgCl$_2$·6H$_2$O, 1 mg/L MnCl$_2$·4H$_2$O, 1.7 mg of ZnCl$_2$, 0.43 mg CuCl$_2$·2H$_2$O, 0.6 mg CoCl$_2$·6H$_2$O, 0.6 mg Na$_2$MoO$_4$·2H$_2$O, pH 6.5) (32) containing 160 mg/L of L-Alanine 7-amido-4-methylcoumarin (AMC) to determine substrate consumption, related to nitrogen and carbon acquisition from peptides, inside the devices. AMC is a fluorogenic substrate that becomes fluorescent (excitation peak at 341 nm and an emission peak at 441 nm) upon cleavage of the amide group by peptidase activity.

The inner part of the microfluidic devices was filled beforehand with the same M9 medium containing 160 mg AMC/L (pH 6.5) by capillary forces directly after bonding or

after reactivation of the chip if filling directly after bonding is not possible. Activation is necessary for the filling of the chip due to the natural hydrophobicity of PDMS. After plasma activation, PDMS remains hydrophilic for some minutes, during which the liquid media has to be pipetted in so that it can be dragged in the channels by capillary forces. The air within the PDMS channels exits through the PDMS, thanks to its gas permeability. The filling process occurs in a period between 10 and 60 seconds. Verification of complete filling of the chip was done by direct observation; filled parts look darker and more transparent than unfilled parts; or using light microscopy, meniscus in the liquid-air interphases can be observed in the chip structures when the chips are not filled. Directly after, a volume of 5 µL of the soil extract was pipetted into the reservoir of the devices. Sterile wet tissues were placed inside the Petri dishes to preserve humidity. The plates were sealed with Parafilm to prevent water from evaporating and kept in the dark at room temperature.

In total, six devices were used for the experiments, three of the channel devices and three of the fractal devices. The experiments were focused mainly on the activity of procaryotes, which are the ones first in the microfluidic devices during the duration of the experiments.

During the run of the experiments, all the studied structures were colonized by microorganisms, confirmed by microscopy. Although there was no filtration process that would leave fungi excluded from the experiment, there were no hyphae observed within the structures during the first 12 days of measurements. In the later stage of the experiment, around day 14, several unicellular eukaryotes were observed to grow in both the pillar system and the experimental structures of the devices. For this reason, data were only analyzed up to day 12 where mainly bacteria, although there could have been other organisms present, affected the measurements. The type of organisms could be identified with light microscopy as bacteria or protist based on their size (between 0.2 and 1 micron) for bacteria and archaea and bigger than 1 micron for protists (ciliates, amoebas, etc.).

## Microscopy

Epifluorescence microscopy was used for visualization of AMC degradation using a fully motorized Nikon Ti2-E inverted microscope with PFS4 hardware autofocus, full 25-mm field-of-view, CoolLED pE300-White MB illumination connected via a 3-mm liquid light guide (LLG), and a Nikon Qi2 camera with 1× F-mount adapter. The filter used was LED-DAPI-A-2360A Semrock Filter Cube (Ex: 380–405 nm, Em: 413–480 nm). Images for fluorescence quantification of the entire device were captured using a (MRH00041) CFI Plan Fluor 4X, N.A. 0.13, W.D. 17.1 mm objective, with an exposure time of 100. NIS-Elements Advanced Research imaging software (Nikon) was used for coordination of the multipoint imaging. Pictures were taken for every device once a day for 12 days. The days selected for statistical analysis were the ones of maximum fluorescence signal for each treatment.

## Image analysis

The fluorescence intensity was quantified using ImageJ 1.52 n (33). Background was subtracted with the ImageJ rolling ball algorithm (34) using seven pixels as radius of rolling ball for 4× objective images. The rolling ball radius was given based on the size of the biggest fluorescent object, which was the diameter of a PDMS channel. After the subtraction, the total intensity was quantified inside each labyrinth using the ROI manager tool. For this, a ROI mask, which contained multiple rectangles of equal size that surrounded each channel or maze, was used to quantify the mean fluorescence intensity within each structure.

For comparison of the fractals of different order, replication consists of the pooled data of the area of each fractal block ($n = 10$, four treatments), which equals the size of five connected modules of the F5 maze lined up after each other. It is inoculated via its connection to the entry pillar system only via the first module (Fig. 1).

## Estimation of accessibility within fractal mazes

The spatial variation of the fluorescence inside the fractal mazes was measured by quantifying the fluorescence at the dead-ends inside the mazes. The spatial analysis was performed on the second module of each fractal block counted from the pillar system, selected to minimize the edge effect produced in the first fractal due to its direct contact with the pillar system. The spatial analysis was done in all internal replicates of one microfluidic device and consisted in comparing the fluorescence in each dead-end to its accessibility index.

The accessibility index was calculated using COMSOL Multiphysics (35). The accessibility index is defined as the time required for a particle in a diffusion simulation to reach 50% of the final concentration and was compared between all dead-end locations ($n = 324$ dead-ends per fractal module) of the maze order F5 or corresponding locations, second module from the pillar system.

This type of measurements was done in dead-ends only so that fluorescence intensity in different regions within the mazes could be systematically compared using regions of interest (ROIs) in ImageJ. Using ROIs allows discretization of the space within the mazes and hence comparison of fluorescence between mazes, maze types, and with their respective accessibility index estimated with COMSOL.

## Statistical analysis

Both experiments with the respective microfluidic device type had full-factorial designs. The channel device experiment had *Angle* (45°, 90°, 109°) and *Turn order* (alternated or repeated) as fixed factors. Each device contained 10 channels of each type (with all the angle-turn order combinations), and three devices were analyzed. Multilevel model fitting correcting for random effects was used to test the influence of every factor on the variables.

A linear regression was performed to test the effect of channel tortuosity in the substrate degradation. Fluorescence intensity of AMC was regressed against the tortuosity of each type of channel considering the microfluidic device as a random variable.

The fractal device had fractal order (F0, F1, F3, F5) as fixed factors, each device had seven mazes of each fractal order type, and three devices were included into the experiment. Multilevel model fitting correcting for random effects was used to test the influence of fractal order in fluorescence intensity. Random effects were attributed to each device as physical replicate of the experiment.

For in-depth statistical analysis in both experiments, the data from the day comprising maximum fluorescence for each fractal maze was chosen. The significance threshold used for all statistical tests was $P < 0.05$. When significant differences were found in the ANOVA, interactions were analyzed separately using the Dunn's method for multiple comparison of means (36). Pairwise comparisons between treatments (channel or maze type) were done with *t*-tests with $P$ values adjusted using Holm corrections (37). All statistical analysis was performed using R (38).

## RESULTS

In device design 1 (channel device), all studied channel types showed a measurable fluorescence signal starting from day 2 of the experiment (Fig. 2 and 3). The fluorescence increased until it reached its maximum for all the channel types at day 5 after inoculation (Fig. 2a). After 5 days, the fluorescence signal decreased until the end of the experiment on day 12.

A comparison of the fluorescence between channel types at the day of maximum signal showed that as channel turning angle increased, the amount of substrate cleaved also increased (Fig. 2b). The fluorescence increased, however, only from 45° angles to 90° angles, after which it did not change significantly, and turn order had no measurable effect. When the tortuosity of the channels was used as the independent variable to

explain the fluorescence patterns in regression against fluorescence intensity, the results suggested a similar output: as tortuosity increased, the amount of cleaved substrate also increased (Fig. 4, $P = 0.046$, $R^2 = 0.017$) (Tables S1 and S2).

The fractal device also showed measurable fluorescence signals in each of its mazes from day 2 after inoculation (Fig. 2 and 5). The maximal fluorescence was found at different time points, delayed with increasing fractal order of the mazes (Fig. 2c); while F0 and F1 had their maximum fluorescent signal on day 2 and 3, respectively, F3 and F5 had it on days 5 and 7, respectively. The comparison between the maximum fluorescence values of each maze showed that as fractal order increased, fluorescent signal also increased (Fig. 2d), highly significantly different from each other except for F0 and F1 (Tables S3 and S4).

A spatial analysis of the fluorescence within the fractal modules indicates that the fluorescence distribution within the mazes depended on the fractal order (Fig. 6). For simple fractals (F0 and F1), the enzymatic activity was higher towards the deeper parts of the maze, but as time passed, the pattern was reverted. Fluorescence in the deeper parts of the maze decreases in comparison to the higher accessible regions. In the more complex fractals (F3 and F1), on the other hand, the pattern was the opposite (Fig. 6).

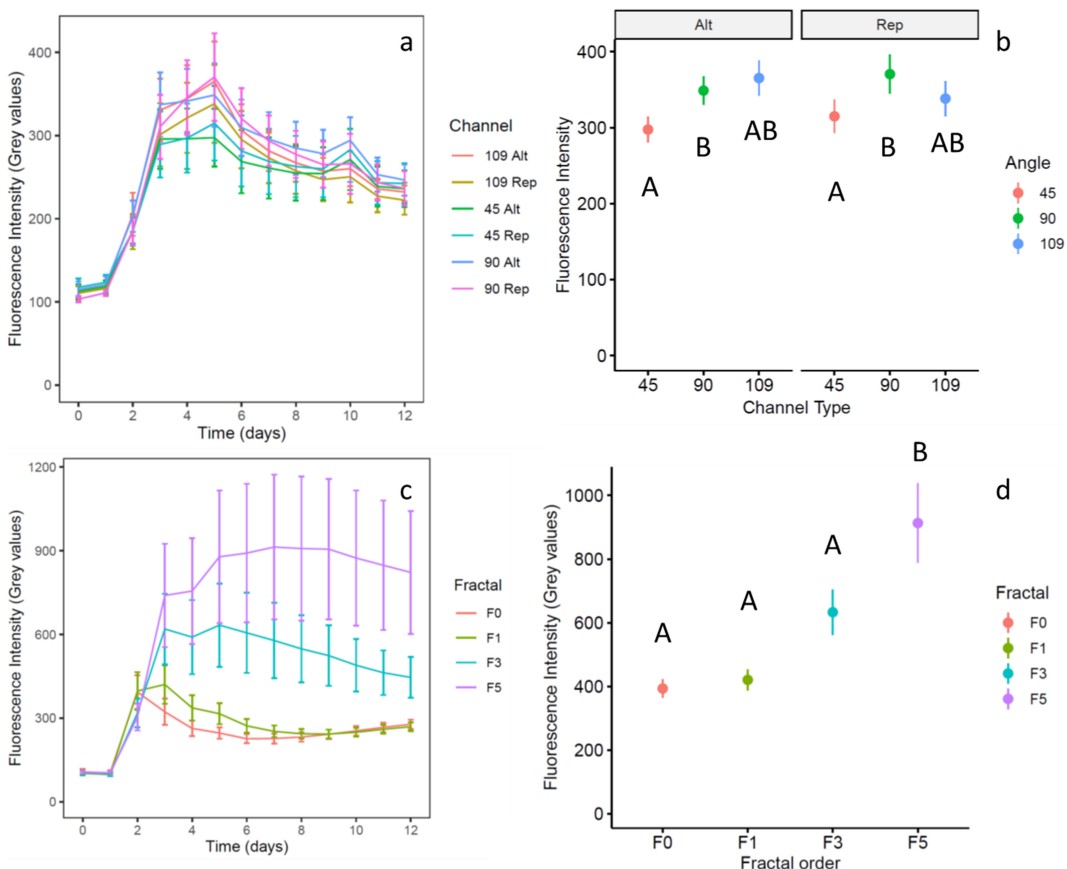

**FIG 2** Fluorescence intensity indicating enzymatic activity in the channel (a and b) and the fractal (c and d) devices over time (a and c) and at the timepoint of maximum fluorescence signal (b and d). The fluorescence data over time for the channel device (a) are shown for the 6 studied treatments (45°, 90°, and 109°, with alternated or repeated turn order each one) with 30 total replicates (10 per device of a total of 3 devices used). The mean fluorescence recorded at the time points of highest mean fluorescence value for each treatment was compared considering fluorescence intensity as dependent variable and angle and turning order as independent variables (b). Fluorescence intensity data over time for the fractal device shown for the four fractal order treatments (fractal order 0, 1, 3, and 5) at normalized volume (c). Comparison of the fluorescence levels in the four maze types of increasing fractal order at the time points of their respective highest mean fluorescence (d). Mean comparisons were done with two- and one-way ANOVA for the channel and the fractal treatment, respectively. Error bars show standard error for $n = 30$ and $n = 21$ for the channel and the fractal condition, respectively. Different capital letters under mean values indicate statistically significant differences between the treatments derived from pairwise comparison using Dunn's method for confident interval adjustment (b and d).

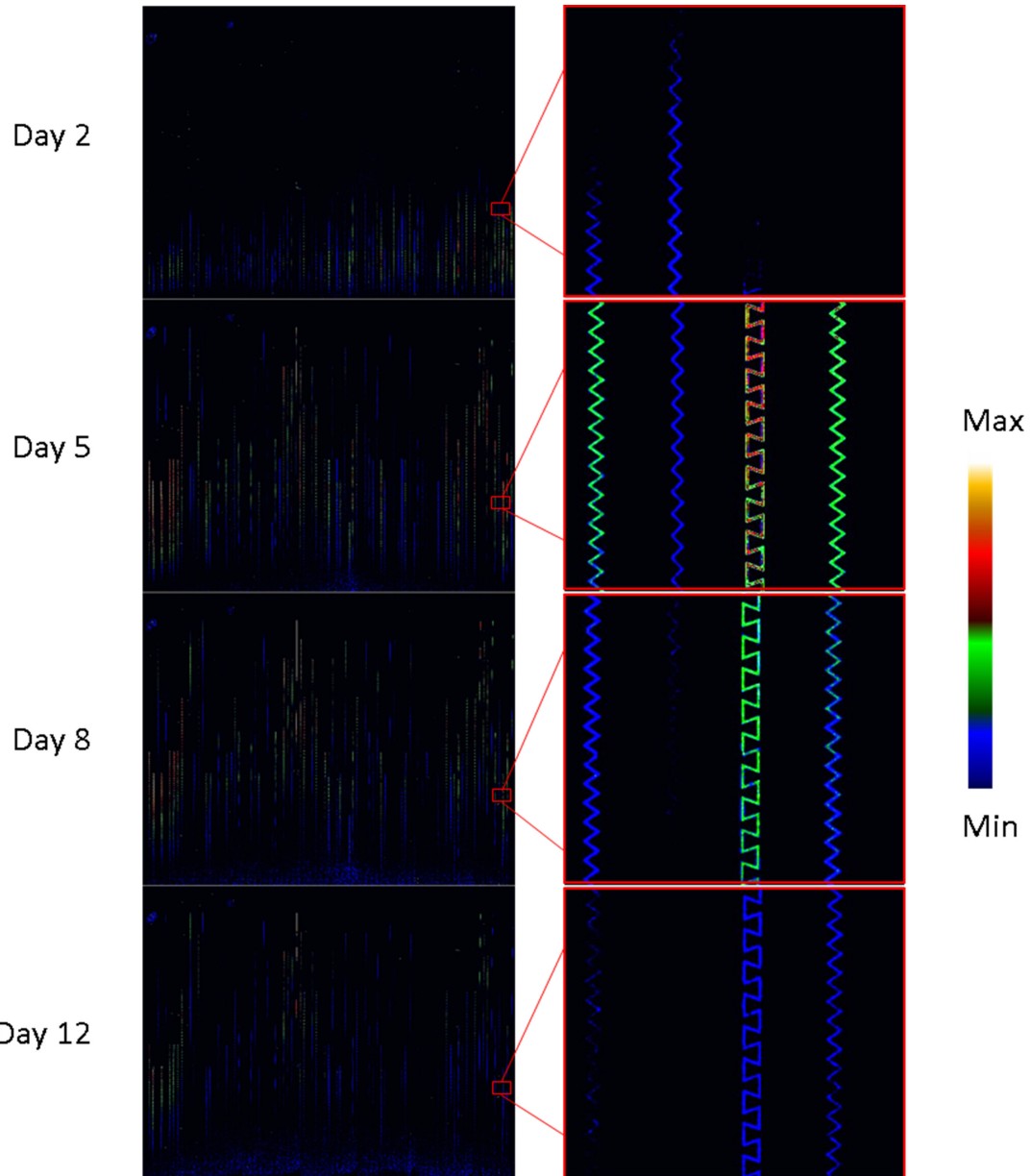

**FIG 3** Example of the fluorescence intensity corresponding to enzymatic activity changing over time in the channel device. Left panel images show the entire device and right panel images show a magnified part of the channel. The fluorescence intensity inside each channel is shown using the color coding placed on the right side of the figure. The selected days correspond to the days after inoculation where the first enzymatic activity is detected (day 2), when it reaches its maximum (day 5), when it starts decreasing (day 8), and when it reaches a lower plateau (day 12).

Fluorescence was higher in the most accessible areas compared to the deeper regions in the earlier stages of the experiment (day 2), but as time passes (from day 3 on), the fluorescence reaches its maximum levels in the deeper regions of the maze. In the most complex fractal, F5, however, this increase was only until a certain point, after which lower fluorescence intensity was located.

Although there is a trend in fluorescence intensity along the different levels of accessibility, there is still a significant part of the variation not explained by accessibility, which suggests the importance of additional factors.

## DISCUSSION

Our hypothesis stated that as spatial accessibility of a microhabitat decreases, the enzymatic activity of a natural inoculum would also decrease. This assumption was made based on previous perspective papers that have proposed the soil spatial microcomplexity as a responsible factor for carbon stabilization (7). The results obtained in this study show, however, the opposite pattern: as simulated habitat accessibility decreases, enzymatic activity increases. In the case of the channels with different turning angles, previous results with single strain showed no effect in the growth of *Pseudomonas putida* (39). However, we observed similar results to the ones in the present work, in a pure culture of the bacterial strain *Pseudomonas putida* in mazes of increasing fractal order where higher fractal order mazes showed more bacterial biomass and nutrient degradation than low fractal order mazes (27).

Previous work showed that when growing in channels with different turning angles, AMC degradation and biomass were unaffected for *Pseudomonas putida* (26), contrary to what occurred in the results of the present work. In the case of the fractal chip, the results were similar, in terms of AMC degradation, to the ones found in *Pseudomonas putida* (27). There are two explanations to the increasing enzymatic activity observed as the spatial fractal order and the turning angle increased: one is that a decrease in habitat accessibility promotes a higher accumulation of bacteria due to quorum sensing, which leads to higher biomass and biofilm formation, which ends up in a higher nutrient acquisition efficiency. Bacterial strains such as *E. coli* have been shown to accumulate in dead-ends of microfluidic mazes due to the action of quorum sensing molecules (40). Quorum sensing can occur not only between bacterial individuals of the same strain, but also between organisms of different species (41), forming what is known

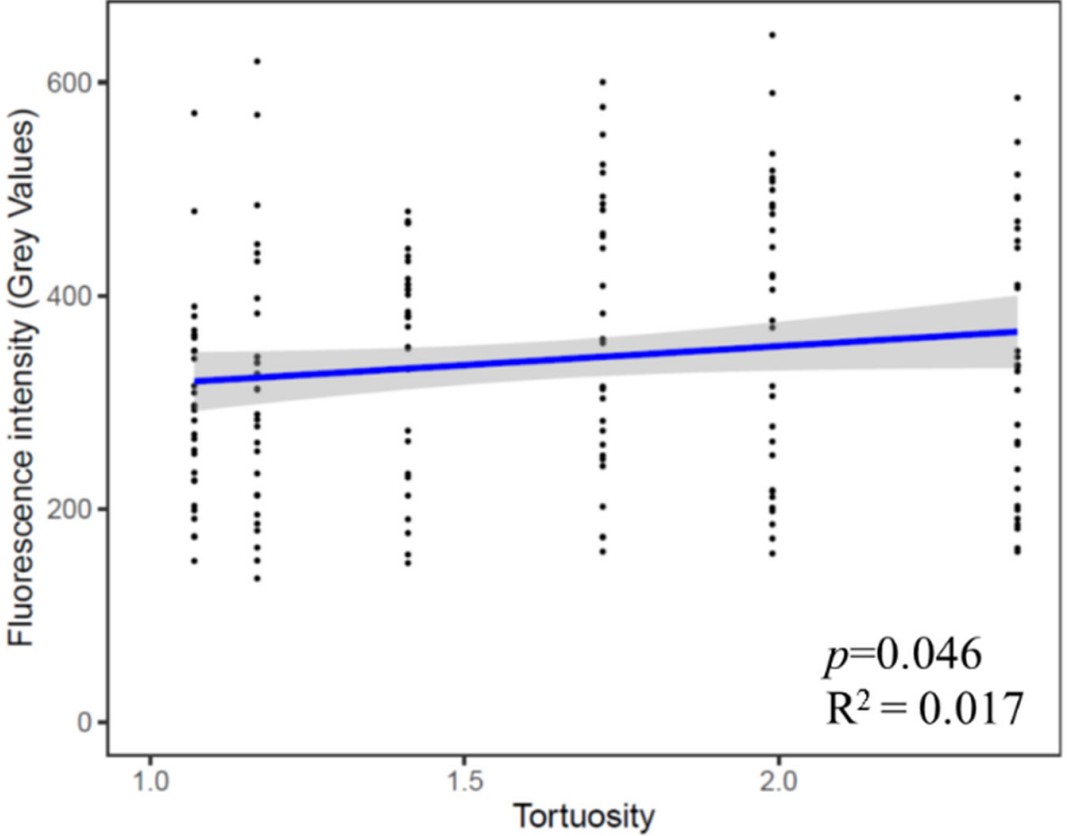

**FIG 4** Linear regression of the fluorescence intensity corresponding to enzymatic activity measured in the different channel types of the channel device as a function of the tortuosity of the channels, at the day of maximum fluorescence for each channel type.

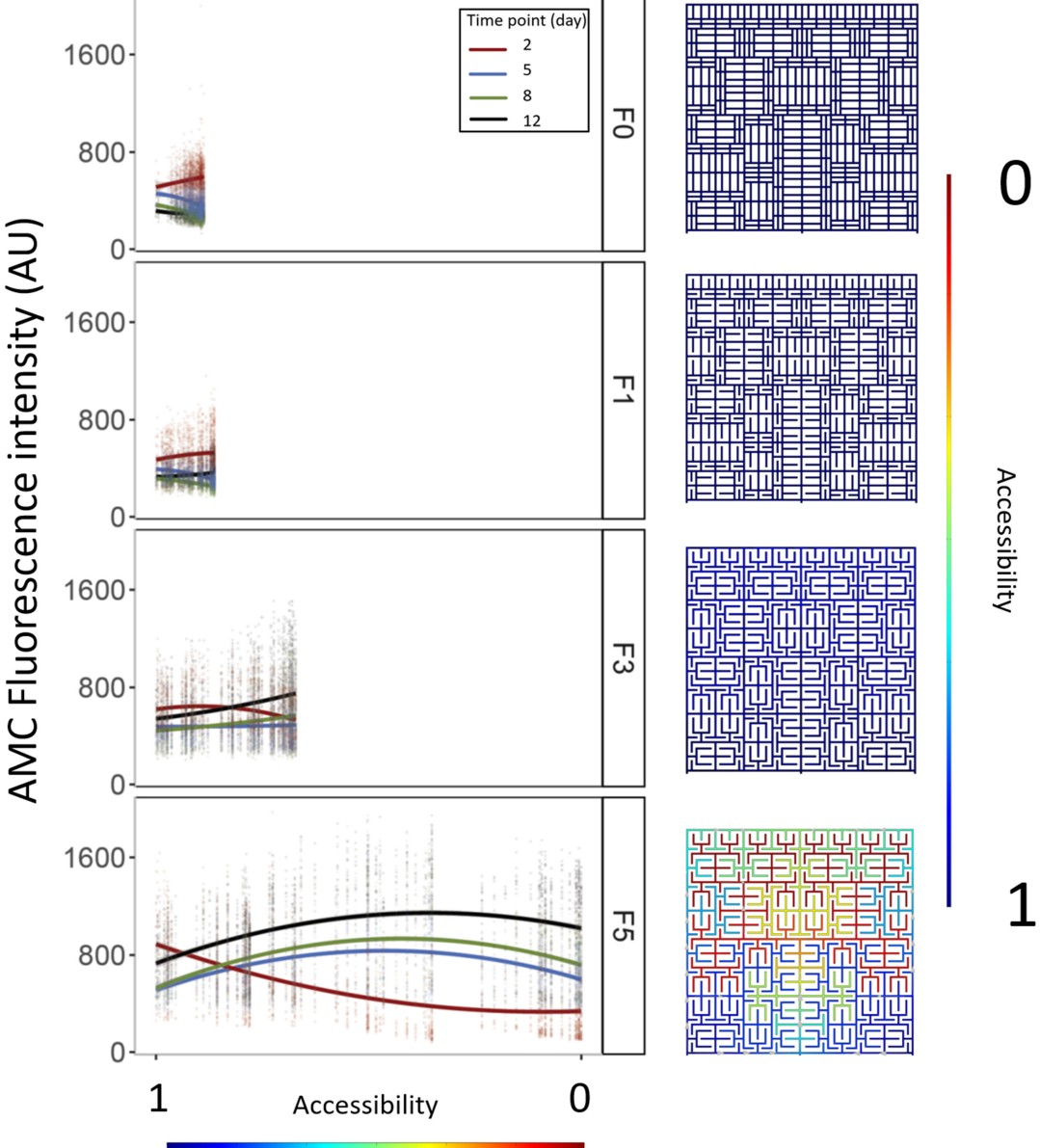

**FIG 5** Spatial distribution of enzymatic activity within the different mazes measured via the fluorescence intensity of AMC. Accessibility of different spaces within the mazes are ranging from 1 to 0, determined via a COMSOL model simulation (right panel), where 1 is the most accessible region and 0 is the least accessible. All fractal order mazes are compared over the whole accessibility range for F5; comparison of the internal maximal variability of accessibility for each fractal order can be appreciated in the shorter curves of the rest of the fractal types compared to the F5 in the figure. Each dot in the regression plots (left panel) denote to the specific spatial accessibility of the COMSOL model and the measured mean fluorescence of that specific region, in all fractal order mazes corresponding to the dead-end locations within F5. The second fractal module of each block was used for analysis of seven internal replicates. The lines correspond to curve fitting using a quadratic model; the colors of the lines and the dots represent the timepoint they correspond to: days 2 (red), 5 (black), 8 (green), and 12 (blue).

as polymicrobial biofilms. These biofilms can have characteristics that differ to a great degree from the sum of characteristics of all the species that conform it (42). One of the main difference is that enzymatic production can be shared between the different species, as exoenzymes can remain active for hours to days, and their products become available for all microbes in the vicinity, becoming in the long run more efficient at degrading substrates via a division of tasks (43).

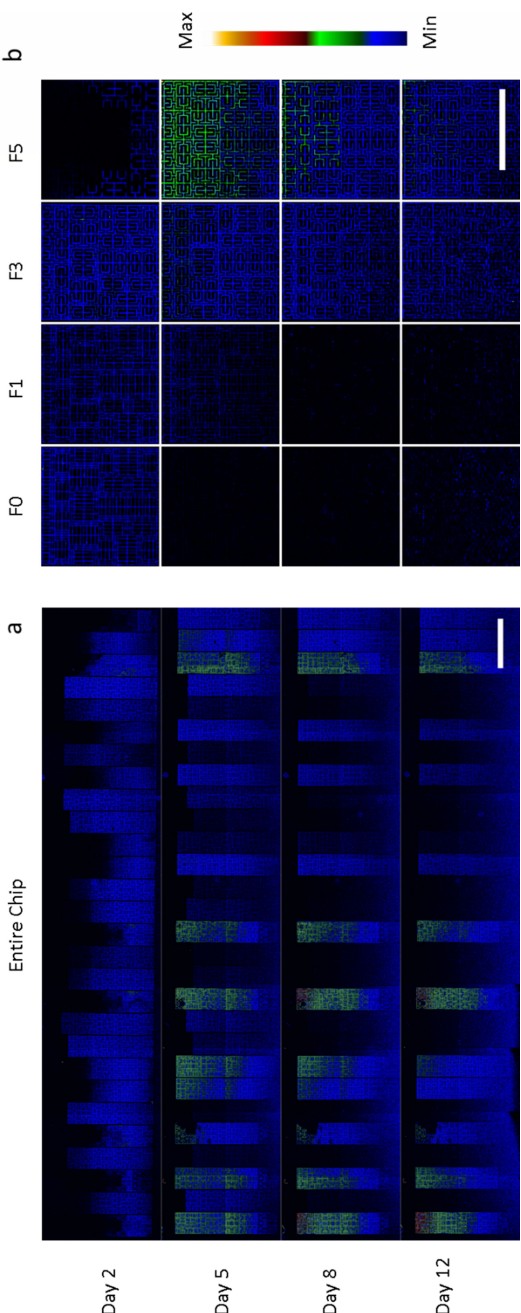

**FIG 6** Example of fluorescence intensities recorded over time in the fractal device. Left panel images show the entire device, and right panel images show a high-magnitude excerpt of each maze type of the second module in each block. The fluorescence intensity within each maze is shown using the color coding to the right of the figure. The selected days correspond to the days after inoculation where the first enzymatic activity is detected (day 2), when it reaches its maximum (day 5), when it starts decreasing (day 8), and when it reaches a lower plateau (day 12).

The mechanisms by which quorum sensing might be enhanced in dead-ends is that molecules involved can accumulate more due to the higher surface area at the dead-ends. This accumulation would then attract more bacteria towards them, which in turn would produce more quorum sensing molecules. Such positive feedback between quorum sensing molecule accumulation and attraction to more bacteria could be one of the explanations why the high-order mazes show a higher enzymatic activity.

A complementary explanation is that a decrease in habitat accessibility reduces the interactions within the bacterial communities and individuals present in the mazes. A reduction of interaction reduces the competition in the different communities, allowing a higher diversity of species and metabolic functions to co-occur (43). This means that species or individuals, which show a preference for the more costly enzymatic acquisition of nutrients, are allowed to grow since potential interactions with fast-growing competitors would be reduced. More connected, easily accessible habitats reduce the fitness of the slow growers and favor communities of fast growers which are in a competitive advantage (44). Since such fast-grower communities have lower tendency to enzymatic nutrient acquisition, this could explain that the enzymatic activity is considerably lower than in less connected environments in our study. Having a physical separation allows otherwise slower growing bacteria to cooperate via sharing metabolic pathways without being outcompeted by opportunists, as it has been seen to happen in pure cultures of bacteria (43).

This is in line with the *species sorting* model that states that the variation of the populations is determined by the environmental characteristics of a particular habitat (45). In a complex habitat, the types of niches are more diverse, allowing several populations to grow with limited interactions with each other, permitting, for instance, coexistence between fast and slow growers. Species sorting has been shown to be responsible of microbial community structures in aquatic environments (46) and might be responsible, at least in part, for the high microbial diversity found in soils. In this sense, trends in the population structure of the community should match those in the habitat structure, which can be partially confirmed with our results: the channels and mazes with the lowest spatial accessibility show a higher enzymatic activity, suggesting that populations exhibiting a higher substrate degradation efficiency have a better fitness than in more spatially homogeneous structures. A low connectivity in the channels and mazes might have created partially disconnected microhabitats where slow growers and fast growers instead of competing cooperate by sharing products of enzymatic activity, or secondary metabolites, leading to a higher substrate degradation efficiency than in fully connected habitats, similar to experiments where spatial separation of competitors promoted coexistence (43). We cannot, however, show if the microbes in our system have a higher substrate degradation efficiency since we cannot, with the methods used in this experiment, determine the microbial biomass for the distinct structures nor determine their diversity. Nonetheless, and based on our previous studies with a bacterial lab strain that showed enzymatic activity uncoupled with the bacterial biomass (27), we can speculate that in our case, we also might have an increased substrate degradation efficiency in complex mazes and channels.

The principal similarity in the results with a single strain in fractal mazes (27) and an entire community might be because habitat complexity might be operating similarly in competition, whether it is interspecific or intraspecific. A reduction in intraspecific competition could allow different metabolic strategies to coexist within the population of a single strain or wider variety of niches being created. Switching to an enzyme production strategy to acquire nutrients could be feasible only if members of the strain that opt for a fast-growing strategy are not in the vicinity. In the same way, in a mixed community such as the one in the present work, slow growers that use enzymes to acquire nutrients would need a low level of competition in order to survive from fast-growing members.

The analysis of changes in fluorescence depending on accessibility indicates that the majority of the enzymatic activity is located in regions with medium level of accessibility. This could be related to the interplay of dead-ends effect, accessibility, and competition. Regions that are easy to access will also have high levels of competition, whereas regions that are difficult to access have lower levels of competition and death ends where quorum sensing molecules could accumulate.

It is interesting to note that in the present experiments, the peak of fluorescent intensity occurs between days 2 and 7 depending on the structure. In previous

experiments with single strains, although the peak in biomass was at around 24 hours, the peak in AMC degradation occurred in later days, around the same time as in the present results (26, 27). This could be due to enzymes that remain cleaving substrate even after cell death, or enzyme-producing members of the community that arise after the easily available nutrients, like glucose, are exhausted. Since the goal of the present study was not to measure biomass or community composition, we can only speculate about these explanations, but this opens up for interesting future studies that look into this parameter. However, knowing how biomass and community composition varies in space and how it correlates with enzymatic activity would clarify the mechanisms behind the patterns we observed in the experiments presented in this work.

The temporal delay or differences in time on the peak in enzymatic activity between mazes (Fig. 2) might be due to the different nutrient use efficiencies of microbes in each maze, which was also observed in experiments with a single bacterial strain (27). While simple mazes (F0 and F1) have a steeper slope in the first day, they reach a maximum that is lower than the one reached in F3 and F5. The community in simple mazes might be dominated by fast growers, which grow at a higher rate from the mineral nutrients and consuming partly the nutrients from the AMC. But since the pace of growth is fast, they run out of mineral nutrients before they can reach higher degradation of the AMC. In complex mazes, on the other hand, growth is slower, mainly dominated by slow growers, we believe, which allows a more efficient use of the mineral resources to invest in the enzymatic degradation of the AMC.

The growth medium used was M9 medium, meaning that all nutrients can be acquired by microbes without the need of extracellular enzymes. The only nutrients obtained via enzymatic activity from the medium used were the nitrogen and carbon of the dipeptide AMC. However, as necromass accumulates in the chip, microbes could also recycle some of the nutrients from it with the use of enzymes. Studying the dynamic of enzymes targeting necromass could therefore be of interest, especially if they are determined by the spatial structure such as the ones needed to cleave the AMC.

Another explanation of our results could be linked to nutrient diffusion. As nutrients are consumed initially in the entrance portion of the chip, the nutrient gradient created, low nutrients in the entrance and high in the mazes or channels, would occasion a nutrient diffusion from the structures to the entrance. And since diffusion happens over shorter distances in simple structures compared to complex ones, complex structures would end up having more nutrients by the time organisms colonize them.

In the present experiments, we tested how spatial variation in a pore space structure affected the activity of peptidases, which presumably was used by the bacteria to complement their nitrogen demand. Other studies in soils have focused on how various enzymes for cleavage of different substrates act around plant roots. It has been shown that the ratios of enzymatic activities are different for each enzyme type, meaning that in regions where carbon specific enzymes are high, those needed to acquire nitrogen or phosphorous are low and vice versa (47). It would thus be interesting to investigate in future studies how the relationship of different enzyme activities is affected by spatial structures in controllable systems, by using a suite of different fluorogenic substrates. Even though our study was mainly focused on prokaryotic nutrient degradation, which is the dominant community in the first 12 days of this study, we know from previous experiments (48) that unicellular eukaryotes also can grow within the devices. Thus, a study on how the onset of predation and trophic food web interactions acts on organic matter substrate consumption in different microstructures can also be a step forward to our study.

We singled out spatial microstructures as the manipulated explanatory factor in our devices, while we had to disregard other variables that likely have a strong influence on OM dynamics in soils, such as the ratio and patchiness of gas/water saturation in the pores, or organo-mineral interactions that may immobilize organic molecules. Under such a decreased accessibility, an increased biological complexity including inter-kingdom interactions likely plays an even more important role, where, e.g., fungi are the

only organism group that easily bridge over air bubbles and can aid the dispersal of other organisms like bacteria and protists, which require water films for movement (48). Nevertheless, our approach was able to demonstrate the considerable effect that the 2-d spatial arrangement of a pore space can have on its microbial colonization and speed of nutrient cycling. This can in the long run lead to a better understanding of the role of pore space characteristics on soil functions and could lead to recommendations for land uses preserving soil structure and their related ecosystem functions.

## AUTHOR AFFILIATIONS

[1]Department of Biology, Lund University, Lund, Sweden
[2]Department of Microbiology & Ecosystem Science, Division of Terrestrial Ecosystem Research, University of Vienna, Vienna, Austria
[3]Department of Biomedical Engineering, Lund University, Lund, Sweden
[4]Centre for Environmental and Climate Science, CEC, Lund University, Lund, Sweden

## AUTHOR ORCIDs

Carlos Arellano-Caicedo (ORCID) http://orcid.org/0000-0002-8048-5559

## FUNDING

| Funder | Grant(s) | Author(s) |
|---|---|---|
| Vetenskapsrådet (VR) | VR-621- 2014-5912 | Edith C. Hammer |
| Stiftelsen för Strategisk Forskning (SSF) | SSF FFL18-0089 | Edith C. Hammer |
| Biodiversity and Ecosystem Services in a Changing Climate | | Edith C. Hammer |

## AUTHOR CONTRIBUTIONS

Carlos Arellano-Caicedo, Conceptualization, Data curation, Formal analysis, Investigation, Methodology, Validation, Visualization, Writing – original draft, Writing – review and editing | Pelle Ohlsson, Conceptualization, Methodology, Supervision, Writing – review and editing | Saleh Moradi, Investigation, Methodology | Edith C. Hammer, Conceptualization, Funding acquisition, Methodology, Project administration, Supervision, Writing – review and editing

## ADDITIONAL FILES

The following material is available online.

### Supplemental Material

**Supplemental tables (Spectrum01898-23-S0001.docx).** Tables S1 to S4.

### Open Peer Review

**PEER REVIEW HISTORY (review-history.pdf).** An accounting of the reviewer comments and feedback.

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
