## [Reviewer comments · Microbiology Spectrum]

Microbiology Spectrum

Microhabitat accessibility determines peptide substrate degradation by soil microbial community

Carlos Arellano-Caicedo, Pelle Ohlsson, Saleh Moradi, and Edith C. Hammer

Corresponding Author(s): Carlos Arellano-Caicedo, Lunds Universitet

Review Timeline:

Submission Date:	September 6, 2023
Editorial Decision:	July 26, 2024
Revision Received:	October 15, 2024
Accepted:	November 12, 2024

Editor: Erik Hom

Reviewer(s): Disclosure of reviewer identity is with reference to reviewer comments included in decision letter(s). The following individuals involved in review of your submission have agreed to reveal their identity: Gao Chen (Reviewer #2)

Transaction Report:

DOI: <https://doi.org/10.1128/spectrum.01898-23>

Re: Spectrum01898-23 (Microhabitat complexity enhances substrate degradation by soil microbial community)

Dear Dr. Carlos Gustavo Arellano-Caicedo:

Thank you for the privilege of reviewing your work. Below you will find my comments, instructions from the Spectrum editorial office, and the reviewer comments.

Sorry for the delay -- I was waiting on another reviewer who failed to submit their review. Nevertheless, I am moving forward and you have reviewers comments to address. Please carefully address them and in your cover letter to me, please briefly summarize your revisions. Further instructions are below.

Revision Guidelines

Sincerely,
Erik Hom
Editor
Microbiology Spectrum

Reviewer #2 (Comments for the Author):

Arellano-Caicedo et al studied carbon degradation with the increase of matric complexity using microfluidics. It is interesting research. The experiments were well performed, and the conclusions were solid. This manuscript is also well written. There are only a few minor comments.

L37, "were" should be "where".

L47, remove the first "in".

L65, "responsible of" should be read as "responsible for".

L344, how can microscopy confirm whether the microorganisms are bacteria or not?

L434, inoculum should be inoculum.

Reviewer #5 (Comments for the Author):

The paper is a logical extension of two other papers from the same team of others using the same kind of microfluidic chips to study spatial arrangement of bacteria and substrate decomposition by bacteria, only this time with natural consortia instead of single strains. The findings are similar. Still I consider this new manuscript to have a right in its own and don't consider it to be incremental gain of knowledge. I see room for improvement along a few different lines:

1. Description of the experiment:

a. Is diffusive oxygen supply only possible through the inlet and could the oxygen concentration therefore scale with accessibility, or is the cover slip permeable to oxygen?

b. How can the micromodel suck up the M9 nutrient solution by capillary forces, if the defending fluid (air?) cannot escape?

c. How did you make sure that no duct was blocked by air entrapment?

2. Spatial analysis:

a. I like the approach to define accessibility by the geodesic distance to the inlet (which is emulated by diffusion distances in this study). However, the scatter in fluorescence intensity for a given accessibility value is still huge (Fig. 5) and I wonder whether this is purely random or triggered by another spatial process. Each crossroad of the maze has a different coordination number (nr. of connecting bonds) or a different length density of accessible paths in their neighborhood (neighborhood could be e.g. a geodesic length of 50 μm), which mimics the amount of available substrate and/or chance of interaction. For instance, dead end paths exist at all accessibility levels, i.e. at all geodesic distances to the inlets in the F3 and F5 model. Could such a second explanatory variable explain some of the observed scatter? Knowledge about this would make a lot of speculation in the discussion section more solid and underpinned by actual data.

3. Interpretation of the findings:

a. Two morphological properties, tortuosity and connectivity/accessibility are framed as being two manifestations of spatial complexity. I'm not a big fan of the term complexity, as it is just not well defined. I would recommend to use habitat connectivity (or habitat accessibility) when referring to the fractal domains.

b. Explanations for increasing enzymatic activity with reduced habitat connectivity: I didn't really get the point why quorum sensing would be increased in a poorly connected network. Sure, the signaling molecules have decreased chances to escape via diffusion and might therefore accumulate, but at the same time the chance that the signal is picked up by other bacteria is also decreased. The second explanation was clearer to me, i.e. the chance of encounter between slow growers with exoenzymes to encounter opportunistic fast growers with a tendency for direct assimilation is smaller, when network connectivity is reduced. However, the only carbon substrate in the M9 nutrient solution was Glucose, right? Glucose is mineralized in soil typically within 2 days, whereas here peak activity is only observed after 3-8 days. So is this peptidase activity which is picked up by the fluorescence signal actually indicative of the recycling of microbial biomass and not of the initial glucose consumption?

L260-261 in the M&M section might therefore be a bit misleading, because AMC cannot determine substrate consumption, if glucose is the substrate. After glucose is fully consumed, opportunistic fast growers should be gone. Therefore, in the later stage that you have addressed, this interaction and competition between fast growers and slow growers should also not play a big role because of the temporal disconnect, right? I don't have a microbiology background, so perhaps I'm oversimplifying. But I do believe that this additional spatial analysis of local accessibility suggested above could help demystifying what's going on.

c. Cross-references to the previous papers by the same authors with single strains in the same microfluidic chips in the discussion section were quite rare and unspecific (Line 440-443, Line 497-501). This felt suspicious, so I read the publication from 2023 completely, only to make sure that we actually learn something new here. Why not play out the strength of comparison and explain in detail if and why natural consortia behaved differently from single strains?

Editorial comments:

L32: remove 'such'

L39: here and elsewhere: consider to replace increasing complexity with decreasing habitat connectivity

L49: consider replacing 'middle region' with intermediate accessibility

L70: I think causation is mixed up. These features, e.g. patch nutrient distribution, give rise to fluctuating habitats

L90: registers -> detects

L91: the process doesn't take place in the matrix, but in the pores

L94: have been detected -> has been demonstrated

L143: These hypotheses are a bit boring in that they reflect what you already know from the previous papers, but I missed a hypothesis about potential differences between single strains and natural consortia

L193: channel dimensions -> channel length

L197-202: some more explanations are required what the characteristic difference between the four models is, e.g. increasing number of blocked pores reduces inherent connectivity and therefore increases mean geodesic distance to the inlet.

L249: remove (full speed)
L260-261: Is AMC turning fluorescent after cleavage? Move info from line 510-511 here.
L300-301: Which regions? Could you do the local accessibility for these regions as outlined above?
L310: Citation looks odd
L312: why only for dead-end locations? Does it mean that you only analyzed dead locations in line 300-301?
L341: Citation looks odd. Isn't it (R Core Team, 2019)?
L366: Is the Sidak method explained in the M&M section?
L381: Change figure order according to occurrence in the text?
L391-396: Can you give an explanation for the temporal delay? I guess it's the change in accessibility, or?
L396-400: Split into two sentences.
L421-430 - Fig. 5: Add color legend for days; Typo in accessibility
L435: decrease -> decrease
L481: goes in line -> is in line
L482: remove all caps in citation
L497: previous our -> our previous
L498: shows -> showed
L510-511: give this info already in the M&M section
L604: Single author: David Hilbert
L656: modify citation
L713: Journal ended up in the title

Dear editor and reviewers,

As authors of the present manuscript, we would like to thank you for taking the time to revise our manuscript and, most of all, for providing invaluable comments that enhanced the clearness and scientific value of our work. We have carefully gone through all your comments, concerns, and suggestions, and have included extra information in the manuscript that we hope would suffice for its publication. The text that has been modified is shown highlighted in yellow in the new version of the manuscript and all changes and responses to each of your comments are addressed in the following document.

Best regards,

Carlos Arellano-Caicedo

Corresponding Author

Answers to reviewers of the manuscript “Microhabitat complexity enhances substrate degradation by soil microbial community”

We would like to thank the reviewers for taking the time of reading of work and for the valuable comments on the experiment, data analysis, and general writing of the document. We found all these comments remarkably useful because they significantly strengthen the relevance and clarity of our findings. All the comments and suggestions have been accepted and are summarize in the present document.

Reviewer #2

Arellano-Caicedo et al studied carbon degradation with the increase of matric complexity using microfluidics. It is interesting research. The experiments were well performed, and the conclusions were solid. This manuscript is also well written. There are only a few minor comments.

L37, "were" should be "where".

Answer from authors: Change has been done in the text.

L47, remove the first "in".

Answer from authors: “in” has been deleted from the text.

L65, "responsible of" should be read as "responsible for".

Answer from authors: Change has been done in the text.

L344, how can microscopy confirm whether the microorganisms are bacteria or not?

Answer from authors: Text has been added where we explain how we can discriminate bacteria from other organisms such as protist based mainly on their size:

Line 278:

“During the run of the experiments, all the studied structures were colonized by microorganisms, confirmed by microscopy. Although there was no filtration process that would leave fungi excluded from the experiment, there were no hyphae observed within

the structures during the first twelve days of measurements. In the later stage of the experiment, around day 14, several unicellular eukaryotes were observed to grow in both the pillar system and the experimental structures of the devices. For this reason, data was only analysed up to day 12 where mainly bacteria, although there could have been other organisms present, affected the measurements. The type of organisms could be identified with light microscopy as bacteria or protist based on their size (between 0.2 – 1 micron) for bacteria and archaea, and bigger than 1 micron for protists (ciliates, amoebas, etc).”

L434, innoculum should be inoculum.

Answer from authors: Change has been done in the text, in line 414.

Reviewer #5:

The paper is a logical extension of two other papers from the same team of others using the same kind of microfluidic chips to study spatial arrangement of bacteria and substrate decomposition by bacteria, only this time with natural consortia instead of single strains. The findings are similar. Still I consider this new manuscript to has a right in its own and don't consider it to be incremental gain of knowledge. I see room for improvement along a few different lines:

1. Description of the experiment:

a. Is diffusive oxygen supply only possible through the inlet and could the oxygen concentration therefore scale with accessibility, or is the cover slip permeable to oxygen?

Answer from authors: The reviewer is right at raising this concern. Oxygen, along other gases, are exchanged with the environment through the reservoir and through the PDMS. There are ways of reducing the diffusivity, using additives in the PDMS, or membranes, but keeping the PDMS at a standard curing agent/elastomer ratio with no membranes allows a constant exchange of gasses. We have included an explanation of this in the methods section, in line 217:

“PDMS is gas permeable and permits the diffusion of oxygen, water vapor, CO₂, and other gasses into the channels (De Bo et al., 2003), meaning that oxygen and other gasses were exchanged with the environment through the chip reservoir and through the PDMS. Making the oxygen not to be a limiting factor within the system.”

b. how can the micromodel suck up the M9 nutrient solution by capillary forces, if the defending fluid (air?) cannot escape?

Answer from authors: This is a true concern when working with designs such as ours. For this reason, we rely on two mechanisms: the surface modification of PDMS through plasma activation, and the gas permeability of the PDMS. PDMS surface is originally hydrophobic, making the filling of the chip challenging for the liquid that comes in contact with the channels is rejected. Hence, surface plasma activation is necessary before filling the chips. The plasma activation modifies the surface properties of the PDMS by replacing temporarily a CH_3 group by an OH group. The chips remain for few minutes hydrophilic, which occasions all liquid that enters in contact with the small channels to be dragged in with capillary forces. Of course, for the liquid to get in, all gases present in the channels need to exit the channels, and since our design is closed, gases can only exit through the PDMS itself. This is possible due to the gas permeability of the PDMS and occurs relatively rapidly, needing around 10 seconds for the entire chip to be completely filled with the liquid medium. This illustrates again that air exchange is not a severe concern in our system.

An extra text in line 259 has been added to clarify this part of the procedure:

“or after re activation of the chip if filling directly after bonding is not possible. Activation is necessary for the filling of the chip due to the natural hydrophobicity of PDMS. After plasma activation, PDMS remains hydrophilic for some minutes, during which the liquid media has to be pipetted in so that it can be dragged in the channels by capillary forces. The air within the PDMS channels exists through the PDMS thanks to its gas permeability. The filling process occurs in a period of time between 10 and 60 seconds.”

c. How did you make sure that no duct was blocked by air entrapment?

Answer from authors: Air entrapment occurs often when filling the chips, especially if the filling is not done directly after plasma activation. This occurs because PDMS rapidly loses hydrophilicity impeding complete saturation of its inner space. To avoid this, the liquid medium needs to be pipetted within a minute after plasma activation. A complete saturation of the chip inner space can be verified by direct observation of the chip structures (with the naked eye), for the liquid-filled space and the air-filled space look different, being the liquid-filled one darker and transparent, and the air-filled one lighter and opaque. It is also possible to verify a complete filling using light microscopy. A chip that is completely filled will show no liquid-air interphase or meniscus within the structures, whereas a chip that is not completely filled has several liquid-air interphases separated by a meniscus.

The following text has been added in line 266:

“Verification of complete filling of the chip was done by direct observation, filled parts look darker and more transparent than unfilled parts; or using light microscopy, meniscus in the liquid-air interphases can be observed in the chip structures when the chips is not completely filled.”

2. Spatial analysis:

a. I like the approach to define accessibility by the geodesic distance to the inlet (which is emulated by diffusion distances in this study). However, the scatter in fluorescence intensity for a given accessibility value is still huge (Fig. 5) and I wonder whether this is purely random or triggered by another spatial process. Each crossroad of the maze has a different coordination number (nr. of connecting bonds) or a different length density of accessible paths in their neighborhood (neighborhood could be e.g. a geodesic length of 50 μm), which mimics the amount of available substrate and/or chance of interaction. For instance, dead end paths exist at all accessibility levels, i.e. at all geodesic distances to the inlets in the F3 and F5 model. Could such a second explanatory variable explain some of the observed scatter? Knowledge about this would make a lot of speculation in the discussion section more solid and underpinned by actual data.

Answer from authors: Indeed, the scatter of fluorescence intensity in space within the fractals is relatively big when plotted against the “accessibility” parameter. This indicates that there are other factors that might be explaining the patterns of enzymatic degradation, such as distance to the entrance, number of connections before the dead end, tortuosity of the different paths to access the dead end, among others. However, these other parameters are implicit in the “accessibility” parameter obtained by the diffusion model. For instance, a dead end with a long, tortuous, and ramified path to the entrance, will have a low accessibility, and one with a short, straight, and unramified path will have a high accessibility. Having a model where all these parameters are considered will produce a complex outcome and little or no profit to the one using the accessibility parameter. Hence, we believe that the variation that is still unexplained by accessibility is due to factors different than the physical one that vary and fluctuate locally, for instance, the type of microbial community present at that specific point, priority effects caused by the order in which different members of the community arrive to the death end, and fluctuation in nutrients or secondary metabolites concentrations, and other parameters such as pH.

Line 509:

“The analysis of changes in fluorescence depending on accesibility indicate that the majority of the enzymatic activity is located in regions with medium level of accesibility. This could be

related to the interplay of dead-ends effect, accessibility, and competition. Regions that are easy to access will also have high levels of competition, whereas regions that are difficult to access have lower levels of competition and death ends where quorum sensing molecules could accumulate.”

3. Interpretation of the findings:

a. Two morphological properties, tortuosity and connectivity/accessibility are framed as being two manifestations of spatial complexity. I'm not a big fan of the term complexity, as it is just not well defined. I would recommend to use habitat connectivity (or habitat accessibility) when referring to the fractal domains.

Answer from authors: We have replaced the term “complexity” when referring to the mazes and channels in the experimental design for “low accessibility”. We also believe that defining the mazes and channels in terms of “accessibility” is less ambiguous than “complexity”.

We have also included the definition of accessibility we use in the manuscript for each chip design:

Line 186:

“In the present work we used the term low accessibility for channels with high turning angle and repeated turning order in the case of the Channel Chip, and mazes with high fractal order in the case of the Fractal Chip. On the other hand, high accessibility was attributed to channels of low turning angle and alternated turning order, and to mazes with low fractal order.”

b. Explanations for increasing enzymatic activity with reduced habitat connectivity: I didn't really get the point why quorum sensing would be increased in a poorly connected network. Sure, the signaling molecules have decreased chances to escape via diffusion and might therefore accumulate, but at the same time the chance that the signal is picked up by other bacteria is also decreased. The second explanation was clearer to me, i.e. the chance of encounter between slow growers with exoenzymes to encounter opportunistic fast growers with a tendency for direct assimilation is smaller, when network connectivity is reduced. However, the only carbon substrate in the M9 nutrient solution was Glucose, right? Glucose is mineralized in soil typically within 2 days, whereas here peak activity is only observed after 3-8 days. So is this peptidase activity which is picked up by the fluorescence signal actually indicative of the recycling of microbial biomass and not of the initial glucose consumption? L260-261 in the

M&M section might therefore be a bit misleading, because AMC cannot determine substrate consumption, if glucose is the substrate. After glucose is fully consumed, opportunistic fast growers should be gone. Therefore, in the later stage that you have addressed, this interaction and competition between fast growers and slow growers should also not play a big role because of the temporal disconnect, right? I don't have a microbiology background, so perhaps I'm oversimplifying. But I do believe that this additional spatial analysis of local accessibility suggested above could help demystifying what's going on.

Answer from authors: The reason is that quorum sensing molecules accumulate in dead ends as it has been shown in previous studies with *E. coli* and modelling, and hence will attract more bacteria to those dead ends, which in turn produce more quorum sensing molecules. We believe that this slowly produces a gradient of quorum signaling with high concentrations at the dead ends, which is amplified over time when more bacteria migrate there.

The reviewer is right when pointing out that glucose is typically respired within 2 days, usually this peak of bacterial growth can be seen between 24 and 48 hours when working with single strains in the microfluidic devices. What we measure in our system in the present set of experiments was the degradation of AMC, which according to our previous results with single strains, is not coordinated in time with maximum bacterial biomass. This suggests two things: that what we observe is the cumulative degradation products of the enzymes produced during the first 24-48 hours, the action of enzymes produced by members of the population, or in the present study of the community that remain active even in stationary or death phase, or both. Since the goal of the present study was not to measure biomass or community composition, we can only speculate about these explanations. However, it would be of great interest to know how biomass and community composition varies in space and how it correlates with substrate degradation.

Although glucose is indeed the primary carbon source, it is not contemplated as substrate in the context of the present work. However, we understand that the term substrate can lead to misunderstandings. For this reason, we have replaced it in the text with “Enzymatic activity” or “peptide substrate”, and have kept the term “substrate” only when talking about substrate degradation efficiency.

If the system had a perfect mix, it would be possible to assume a clear population dynamic where slow growers become abundant later in time. However, since the microenvironment where they grew in the present set of experiments has different levels of connectivity, the competition between fast growers and slow growers is likely to be time and location dependent within the mazes. In fact, the reason why the high order fractal mazes could have such a high enzymatic degradation could indicate a synergy

between fast and slow growers where both benefit instead of outcompeting each other. This is a speculative explanation of the results but could be an interesting starting hypothesis for future studies focusing on competition and cooperation at the microscale.

We found all the comments mentioned above very helpful to delve deeper into the patterns we see in our results and hence have including the following text in the revised version of the manuscript:

Line 447:

“The mechanisms by which quorum sensing might be enhanced in dead ends is that molecules involved can accumulate more due to the higher surface area at the dead ends. This accumulation would then attract more bacteria towards them, which in turn would produce more quorum sensing molecules. Such positive feedback between quorum sensing molecule accumulation, and attraction to more bacteria, could be one of the explanations why the high order mazes show a higher enzymatic activity.”

Line 516:

“It is interesting to note that in the present experiments the peak of fluorescent intensity occurs between day 2 and 7 depending on the structure. In previous experiments with single strains, although the peak in biomass was at around 24 hours, the peak in AMC degradation occurred in later days , around the same time as in the present results (Arellano-Caicedo et al., 2021, 2023). This could be due to enzymes that remain cleaving substrate even after cell death, or enzyme producing members of the community that arise after the easily available nutrients, like glucose, are exhausted. Since the goal of the present study was not to measure biomass or community composition, we can only speculate about these explanations, but this opens up for interesting future studies that look into this parameter. However, knowing how biomass and community composition varies in space and how it correlates with enzymatic activity would clarify the mechanisms behind the patterns we observe in the experiments presented in this work.”

Line 544:

“The growth medium used was M9 medium, meaning that all nutrients can be acquired by microbes without the need of extracellular enzymes. The only nutrients obtained via enzymatic activity from the medium used was the nitrogen and carbon of the dipeptide AMC. However, as necromass accumulates in the chip, microbes could also recycle some of the nutrients from it with the use of enzymes. Studying the dynamic of enzymes targeting necromass could therefore be of interest, especially if they are determined by the spatial structure such as the ones needed to cleave the AMC.”

Line 483:

“A low connectivity in the channels and mazes might have created partially disconnected microhabitats where slow growers and fast growers instead of competing, cooperate by sharing products of enzymatic activity, or secondary metabolites, leading to a higher substrate degradation efficiency than in fully connected habitats, similar to experiments where spatial separation of competitors promoted coexistence (Wu 2016).”

c. Cross-references to the previous papers by the same authors with single strains in the same microfluidic chips in the discussion section were quite rare and unspecific (Line 440-443, Line 497-501). This felt suspicious, so I read the publication from 2023 completely, only to make sure that we actually learn something new here. Why not play out the strength of comparison and explain in detail if and why natural consortia behaved differently from single strains?

Answer from authors: This is a fair point since it is curious that a single strain shows the same tendency in the result as an entire community in fractal mazes, and different in channels with varying turning angles. The reason why we think this might be happening is due to a reduction in competition which can occur at the single strain (intraspecific competition) and between strains in a community (interspecific competition). A reduction in intraspecific competition could allow different metabolic strategies to coexist within the population of a single strain, in our case, the ones related to enzymatic acquisition of nutrients and the acquisition of nutrients in mineral form. Switching to an enzyme production strategy to acquire nutrients could be feasible only if members of the strain that opt for a fast-growing strategy are not in the vicinity. In the same way, in a mixed community, slow growers that use enzymes to acquire nutrients would need a low level of competition in order to survive from fast growing members.

Another difference we found between our previous publication is that when growing in channels with different turning angles AMC degradation was unaffected for *Pseudomonas putida*. On the other hand, turning angle alone showed higher enzymatic activity when using a soil inoculum. The reason, we believe, might be due to the effect swimming patterns have on the colonization of this type of channels. While *Pseudomonas putida* have a similar swimming pattern that permits a homogeneous colonization of the channels, hence not allowing for enough segregation for limiting competition, the soil inoculum contains presumably several strains capable of swimming and differing in swimming patterns. This would mean that depending on the swimming patterns bacteria will position along the channels reducing hence their direct competition. Although this needs to be tested using several strains that differ in

swimming patterns as well as in metabolic strategies to be confirmed, it already shows the different each physical parameter of the microenvironment affects a single strain and an entire community.

We have included the following text in the new version of the manuscript:

Line 426:

“Previous work showed that when growing in channels with different turning angles, AMC degradation and biomass was unaffected for *Pseudomonas putida* (Arellano-Caicedo et al., 2021), contrary to what occurred in the results of the present work.”

Line 498:

“The principal similarity in the results with a single strain in fractal mazes (Arellano-Caicedo et al., 2023), and an entire community might be because habitat complexity might be operating similarly in competition, whether it is interspecific, or intraspecific. A reduction in intraspecific competition could allow different metabolic strategies to coexist within the population of a single strain or wider variety of niches being created. Switching to an enzyme production strategy to acquire nutrients could be feasible only if members of the strain that opt for a fast-growing strategy are not in the vicinity. In the same way, in a mixed community such as the one in the present work, slow growers that use enzymes to acquire nutrients would need a low level of competition in order to survive from fast growing members.”

Editorial comments:

L32: remove 'such'

Answer from authors: “such” has been removed from the specified line.

L39: here and elsewhere: consider to replace increasing complexity with decreasing habitat connectivity

Answer from authors: We have replaced complexity with decreasing habitat connectivity as suggested.

L49: consider replacing 'middle region' with intermediate accessibility

Answer from authors: We have replaced “middle region” with “intermediate accessibility” in line 47.

L70: I think causation is mixed up. These features, e.g. patch nutrient distribution, give rise to fluctuating habitats

Answer from authors: We have replaced: “In soils, a spatiotemporal fluctuating habitat gives to a patchy nutrient distribution...” with “In soils, spatiotemporal fluctuating patchy nutrient distribution gives rise to diverse array of habitats...” in line 69.

L90: registers -> detects

Answer from authors: “registers” has been replaced by “detects”

L91: the process doesn't take place in the matrix, but in the pores

Answer from authors: “matrix” has been replaced with “porous system”

L94: have been detected -> has been demonstrated

Answer from authors: “have been detected” has been replaced with “has been demonstrated”

L143: These hypotheses are a bit boring in that they reflect what you already know from the previous papers, but I missed a hypothesis about potential differences between single strains and natural consortia

Answer from authors: The patterns in our previous results with a single strain of *Pseudomonas putida* showed no effect of channel turning angle on substrate consumption, whereas fractal order showed an increase substrate consumption. We attributed this to the fact that channels in a saturated system, with the studied angles, are not sufficient by themselves to produce a heterogeneous bacterial distribution and substrate consumption, whereas fractal order does. We believe fractal mazes of high order promoted cooperation between individuals, due to a reduced intraspecific competition, as well as favorable conditions to quorum sensing molecules accumulation. Hence, we expected cooperation in natural microbial consortia as well as coexistence of a variety of metabolic strategies in mazes with low connectivity, leading to high enzymatic degradation. On the other hand, we expected to see no effect of the turning angle and order in enzymatic degradation as it was seen in previous experiments with a single strain that these factors did not suffice to produce different AMC signals.

The following text has been added:

Line 134:

“In a single strain experiment, it was found that channels with sharp turning angle did not affect the enzymatic activity of a single bacterial strain (Arellano-Caicedo et al. 2021). On the other hand, mazes with low connectivity led to higher enzymatic activity (Arellano-Caicedo et al., 2023). This could be the product of cooperation mediated via quorum sensing or to a reduction in intraspecific competition that mazes but not channels promote. However, we believed this might not be reflecting what occurs in nature with complex soil communities. Hence, we expected cooperation in natural microbial consortia to be lower than in a single strain since competition would be higher for nutrients and space. We hypothesized that this would lead to higher substrate enzymatic activity in channels with low turning angle and in simple mazes compared to channels with high turning angle and complex mazes respectively”.

L196: channel dimensions -> channel length

Answer from authors: “channel dimensions” has been changed to “channels lengths”

L197-202: some more explanations are required what the characteristic difference between the four models is, e.g increasing number of blocked pores reduces inherent connectivity and therefore increases mean geodesic distance to the inlet.

Answer from authors: We agree that an explanation of the parameters that change with each fractal order maze is necessary. Hence, we have included the following text in line 182:

“With increasing fractal order accessibility and connectivity in the mazes decreases, and tortuosity and mean geodesic distance to the inlet increases.”

L249: remove (full speed)

Answer from authors: “full speed” has been removed

L260-261: Is AMC turning fluorescent after cleavage? Move info from line 510-511 here.

Answer from authors: The following text has been added to clarify the concept and usage of AMC in line 254:

“AMC is a fluorogenic substrate that becomes fluorescent (excitation peak at 341 nm and an emission peak at 441 nm) upon cleavage of its amide group by peptidase activity.”

L300-301: Which regions? Could you do the local accessibility for these regions as outlined above?

Answer from authors: The part of the spatial variation of fluorescence in fractal mazes has been put all under “Estimation of accessibility within Fractal Mazes” in Line 321 and it now reads thus:

“The spatial variation of the fluorescence inside the fractal mazes was measured by quantifying the fluorescence at the dead-ends inside the mazes. The spatial analysis was performed on the second module of each fractal block counted from the pillar system, selected to minimize the edge effect produced in the first fractal due to its direct contact with the pillar system. The spatial analysis was done in all internal replicates of one microfluidic device and consisted in comparing the fluorescence in each dead end to its accessibility index.

The accessibility index was calculated using COMSOL Multiphysics® (Multiphysics & Multiphysics, 2020). The accessibility index is defined as the time required for a particle in a diffusion simulation to reach 50% of the final concentration and was compared between all dead-end locations (n=324 dead ends per fractal module) of the maze order F5 or corresponding locations, second module from the pillar system.”

L310: Citation looks odd

Answer from authors: Citation has been fixed.

L312: why only for dead-end locations? Does it mean that you only analyzed dead locations in line 300-301?

Answer from authors: We did the measurements in dead-ends only because we wanted to be able to compare systematically fluorescence intensity in different regions within the mazes using regions of interest (ROIs) in ImageJ. Using ROIs allows discretization of the space within the mazes and hence comparison of fluorescence between mazes, maze types, and with their respective accessibility index estimated with COMSOL. To clarify this, we have included the following text in the manuscript in line 335:

“This type of measurements was done in dead-ends only so that fluorescence intensity in different regions within the mazes could be systematically compared using regions of interest (ROIs) in ImageJ. Using ROIs allows discretization of the space within the mazes and hence comparison of fluorescence between mazes, maze types, and with their respective accessibility index estimated with COMSOL.”

L341: Citation looks odd. Isn't it (R Core Team, 2019)?

Answer from authors: Citation has been fixed in line 366.

L366: Is the Sidak method explained in the M&M section?

Answer from authors: The correction used in the multiple comparisons in this work was the Dunn's correction for multiple comparisons. The text that contained Sidak correction in the manuscript was replaced by Dunn's correction which is mentioned in line 363.

L381: Change figure order according to occurrence in the text?

Answer from authors: The figures have been placed according to their order in the text.

L391-396: Can you give an explanation for the temporal delay? I guess it's the change in accessibility, or?

Answer from authors: This temporal delay or differences in time on the peak in substrate degradation is, we believe, due to the different nutrient use efficiencies of microbes in each maze. While simple mazes (F0 and F1) have a steeper slope in the first day, they reach a maximum that is lower than the one reached in F3 and F5. The community in simple mazes might be dominated by fast growers, which grow at a higher rate from the mineral nutrients provided in the medium and consuming partly the nutrients from the AMC. But since the pace of growth is fast they run out of mineral nutrients before they can reach higher degradation of the AMC. In complex mazes, on the other hand, growth is slower, mainly dominated by slow growers, we believe, but it allows for a more efficient use of the mineral resources to invest in the enzymatic degradation of the AMC. Hence, the result is a faster but lower AMC degradation in simple fractals, and a slower but higher degradation in high order fractals.

The following text has been added to the manuscript to include this explanation on fluorescent delay in time between mazes:

Line 531:

“The temporal delay or differences in time on the peak in enzymatic activity between mazes (Figure 2) might be due to the different nutrient use efficiencies of microbes in each maze, which was also observed in experiments with a single bacterial strain (Arellano-Caicedo et al., 2023). While simple mazes (F0 and F1) have a steeper slope in the first day, they reach a maximum that is lower than the one reached in F3 and F5. The community in simple mazes might be dominated by fast growers, which grow at a

higher rate from the mineral nutrients and consuming partly the nutrients from the AMC. But since the pace of growth is fast, they run out of mineral nutrients before they can reach higher degradation of the AMC. In complex mazes, on the other hand, growth is slower, mainly dominated by slow growers, we believe, which allows a more efficient use of the mineral resources to invest in the enzymatic degradation of the AMC.”

L396-400: Split into two sentences.

Answer from authors: The sentence in the indicated lines has been separated into two. Now it reads as follows:

Line 396:

“For simple fractals (F0 and F1), the enzymatic activity was higher towards the deeper parts of the maze, but as time passed the pattern was reverted. Fluorescence in the deeper parts of the maze decreases in comparison to the higher accessible regions.”

L421-430 - Fig. 5: Add color legend for days; Typo in accessibility

Answer from authors: The color legends for days have been added to the image and the typo corrected.

L435: decrease -> decrease

Answer from authors: The typo has been corrected in the manuscript.

L481: goes in line -> is in line

Answer from authors: Correction has been done in the manuscript.

L482: remove all caps in citation

Answer from authors: The citations that contained all caps were corrected.

L497: previous our -> our previous

Answer from authors: The correction has been made in the text.

L498: shows -> showed

Answer from authors: The correction has been done in the manuscript.

L510-511: give this info already in the M&M section

Answer from authors: Two texts have been added to include info from Line 510-511 into the Materials and Methods section:

Line 252:

“...containing 160 mg/L of L-Alanine 7-amido-4-methylcoumarin (AMC) to determine substrate consumption, related to Nitrogen and Carbon acquisition from peptides, inside the devices.”

Line 275:

“The experiments were focused mainly on the activity of procaryotes, which are the ones first in the microfluidic devices during the duration of the experiments.”

L604: Single author: David Hilbert

Answer from authors: The reference has been corrected to a single author.

L656: modify citation

Answer from authors: The citation has been corrected.

L713: Journal ended up in the title

Answer from authors: The correction has been done to the reference.

Re: Spectrum01898-23R1 (Microhabitat accessibility determines peptide substrate degradation by soil microbial community)

Dear Dr. Carlos Gustavo Arellano-Caicedo:

Your manuscript has been accepted, and I am forwarding it to the ASM production staff for publication.

Please make sure to make the edits/corrections in production stage that are suggested by the Reviewer (see below). I do not want to hold up an "accept" decision on those minor points!

Your paper will first be checked to make sure all elements meet the technical requirements. ASM staff will contact you if anything needs to be revised before copyediting and production can begin. Otherwise, you will be notified when your proofs are ready to be viewed.

Sincerely,
Erik Hom
Editor
Microbiology Spectrum

Reviewer #5 (Comments for the Author):

The authors have done a good job in responding to my earlier comments and revising the MS accordingly. Please do another careful proof-reading during production, especially of the revised text:

L138: be reflecting -> reflect

L220: Sentence incomplete?

L264: exists -> exits

L269: is -> are

L270: were -> was

...